# EG-ENAS: Efficient and Generalizable Evolutionary Neural Architecture Search for Image Classification

Mateo Avila Pava[1]  René Groh[1]  Andreas M. Kist[1]

[1]Friedrich-Alexander-Universität Erlangen-Nürnberg

**Abstract**  Neural Architecture Search (NAS) has become a powerful method for automating the design of deep neural networks in various applications. Among the different optimization techniques, evolutionary approaches stand out for their flexibility, robustness, and capacity to explore diverse solutions. However, evaluating neural architectures typically requires training, making NAS resource-intensive and time-consuming. Additionally, many NAS methods lack generalizability, as they are often tested only on a small set of benchmark datasets. To address these two challenges, we propose a new efficient NAS framework based on evolutionary computation, which reuses available pretrained weights and uses proxies to reduce redundant computations. We initially selected a reduced RegNetY search space and incorporated architectural improvements and regularization techniques for training. We developed a dataset-aware augmentation selection method to efficiently identify the best transform for each dataset using zero-cost proxies. Additionally, we propose a ranking regressor to filter low-potential models during initial population sampling. To reduce training time, we introduce a weight-sharing strategy for RegNets that reuses pretrained stages and transfers the stem from parent to child models across generations. Experimental results show that our low-cost (T0) and full EG-ENAS (T6) configurations consistently achieve robust performance across eleven datasets, outperforming Random Search (T1) and simple Evolutionary NAS (T2) with competitive results in under a 24-hour time budget on seven validation datasets. We achieve state-of-the-art accuracy on one and surpass the 2023 Unseen NAS Challenge top scores on four datasets. The code is available at this link.

## 1 Introduction

Recent advancements in Artificial Intelligence (AI) have been driven largely by the growing sophistication of neural networks. As architectural design choices rapidly diversify, manually crafting neural network architectures has become increasingly complex, highlighting the need for automated methods, such as neural architecture search (NAS). NAS explores a vast space of potential network structures and optimizes for performance metrics such as accuracy, efficiency, and computational cost to select the best architecture for a given task or dataset [1]. Besides reinforcement learning [2, 3], Bayesian optimization [4, 5], and gradient-based optimization [6], evolutionary algorithms (EAs) are commonly used as optimization methods in NAS [7, 8]. Inspired by natural selection, EAs balance exploration and exploitation through genetic operators like mutation and crossover, enabling the discovery of diverse and novel architectures.

Evolutionary NAS (ENAS), while powerful, often requires training multiple models, leading to high computational costs. Additionally, the stochastic nature of genetic operators can result in extensive exploration of suboptimal solutions [1] if not properly addressed. Many NAS methods also suffer from limited generalization across datasets, as they are typically tested on similar benchmarks and homogeneous datasets. Ensuring better generalizability strengths NAS robustness to unseen data distributions, while improving efficiency reduces computational costs and energy consumption. These advancements would support and enable the adoption of NAS in diverse fields, while also opening the door for more researchers and institutions, including those with limited resources, to access and benefit from NAS.

Building on this motivation, we aim to evaluate the challenges to generalizability and efficiency of each component of ENAS. We also propose an ENAS pipeline applicable to diverse image classification datasets while minimizing both time and computational resource requirements. Recent advances in weight-sharing methods, proxy-based approaches, and surrogate models have demonstrated favorable trade-offs between accuracy and computational cost [9]. Consequently, we incorporate these strategies into our pipeline wherever applicable. We summarize our contributions as follows:

1. Introduce an innovative, zero-cost proxy-based, dataset-aware augmentation selection method that efficiently selects the best transform from a list of 22 options in just a few minutes. This method is applicable to both image and non-image data.

2. Improve the architecture, training, and regularization techniques of networks within the RegNetY design space, as well as its utilization as our search space.

3. Use of a ranking regressor to select the most promising architectures to achieve good accuracy, which will serve as the initial population in ENAS.

4. Introduce a stage transfer and inheritance method for the RegNetY search space to reduce training time during the fitness evaluation step.

Rather than relying only on commonly used image classification datasets like ImageNet16-120 [10] or CIFAR10 [11], we test our approach on nine diverse datasets. These datasets, shown in Figure 2, were used in the NAS Unseen-Data challenge 2023 and 2024 [12], and include tasks where either simple algorithms outperform naive deep learning models or problems too complex for humans without specialized tools. The diversity and challenging nature of these datasets provide a robust framework for evaluating the generalizability of NAS methods.

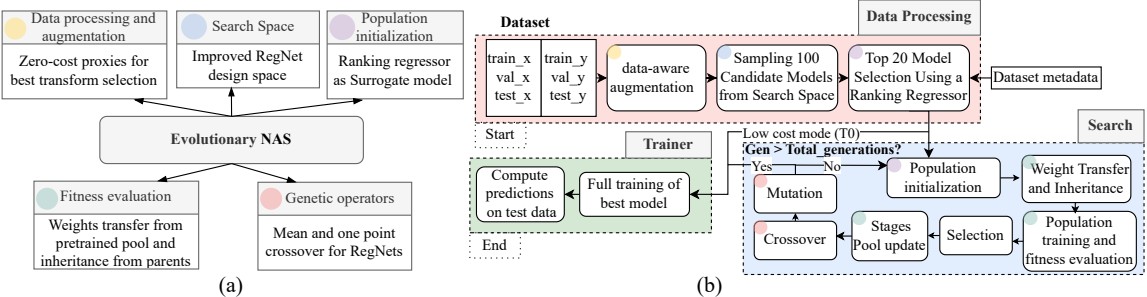

Figure 1: **EG-ENAS Overview** - (a) Our main contributions to EG-ENAS. (b) Pipeline of our EG-ENAS, based on the modules of the NAS Unseen Data Challenges. It begins with the data processing module, followed by the search module, and finally the trainer model. We include an alternative low-cost mode (T0) that uses a surrogate model to find the best architecture, bypassing the search module.

## 2 Background

**Evolutionary NAS (ENAS).** ENAS is an optimization method inspired by biological evolution that uses genetic algorithms to refine neural network architectures through genetic operators, such as crossover or mutation. While evolutionary approaches have been used for over 30 years, their popularity has surged in the last decade due to advances in technology. This progress has led to strong results, e.g., AmoebaNet [22], which was one of the first architectures developed using ENAS (regularized evolution [23]) to achieve high performance in CNN-based classification tasks. ENAS iteratively evolves candidate architectures, selecting the best-performing models to generate new architectures via two key operators: crossover, which combines elements from

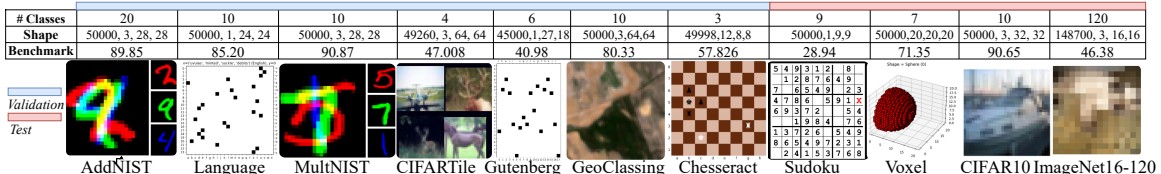

| # Classes | 20 | 10 | 10 | 4 | 6 | 10 | 3 | 9 | 7 | 10 | 120 |
|---|---|---|---|---|---|---|---|---|---|---|---|
| Shape | 50000, 3, 28, 28 | 50000, 1, 24, 24 | 50000, 3, 28, 28 | 49260, 3, 64, 64 | 45000,1,27,18 | 50000,3,64,64 | 49998,12,8,8 | 50000,1,9,9 | 50000,20,20,20 | 50000, 3, 32, 32 | 148700, 3, 16,16 |
| Benchmark | 89.85 | 85.20 | 90.87 | 47.008 | 40.98 | 80.33 | 57.826 | 28.94 | 71.35 | 90.65 | 46.38 |
|  | AddNIST | Language | MultNIST | CIFARTile | Gutenberg | GeoClassing | Chesseract | Sudoku | Voxel | CIFAR10 | ImageNet16-120 |

Figure 2: **Overview of validation/testing split at dataset-level** - Validation datasets (blue) were used to assess EG-ENAS components and were part of the NAS Unseen Data Challenge 2023 [13, 14, 15, 16, 17, 18, 19]. Test datasets (red) include two datasets from the NAS Unseen Data Challenge 2024 [20, 21] and two standard NAS benchmarks, CIFAR10 [11] and ImageNet16-120 [10]. Benchmark scores are available in the NAS Challenge repository [12], except for Voxel and Sudoku, for which we used a ResNet18 model.

parent architectures, and mutation, which introduces random modifications. This approach offers advantages over gradient-based methods by enabling flexible exploration of architectural variations without requiring differentiability. However, its high computational cost remains a challenge, as evolving networks across many generations demand significant resources.

**Efficiency and Generalizability in NAS**. Various techniques have been developed to reduce the computational cost of searching for optimal architectures. Weight-sharing approaches, such as Differentiable Architecture Search (DARTS) [24] and Efficient NAS [25], significantly reduce search time by reusing shared parameters across candidate architectures instead of training each one from scratch. Zero-cost proxies [26] and low-fidelity approximations [27] further accelerate the search by providing quick performance estimates. While these methods improve efficiency, ensuring that NAS-discovered architectures generalize across different datasets and tasks remains an open challenge. Some works have proposed generalizability and transferability methods to improve the robustness of NAS [28, 29, 30]. However, in image classification, NAS methods are often evaluated on homogeneous datasets, such as CIFAR-10, ImageNet, or similar benchmarks, limiting their ability to generalize to diverse real-world scenarios.

**Surrogate models and Zero-cost proxies**. Surrogate models are computationally efficient approximations used to estimate the performance of complex systems without requiring full evaluations. In NAS, surrogate models are often employed to predict model accuracy or efficiency based on architectural features, reducing the need for costly training and evaluation [31, 32]. A more recent alternative to surrogate models in NAS is zero-cost proxies, which estimate model quality without training based on intrinsic properties of the architecture. These proxies are based on analytical heuristics such as gradient sensitivity, parameter saliency, or Jacobian-based metrics. Some of the main zero-cost proxy solutions include SynFlow [33], SNIP [34], GraSP [35], Jacov [36], and Fisher information [37]. These techniques provide rapid evaluations, making NAS more scalable, although their accuracy in ranking architectures remains an area of active research.

## 3 Search space and Evolutionary operators

Selecting an appropriate search space is crucial for the efficiency and effectiveness of NAS. A larger search space offers more diverse architectures but increases search time, while a constrained space may limit the discovery of optimal models. To address this, we adopt the RegNet design space [38], which provides simple and efficient networks that perform well across different FLOP regimes and include a high concentration of top-performing architectures. They outperform EfficientNet under comparable settings, run up to five times faster on GPUs, and surpass standard ResNe(X)t models. RegNetX space contains $10^7$ possible architectures, with their stage widths and depths determined by a quantized linear function, as illustrated in Figure 3.

RegNetX is parameterized by six values: $d, w_0, w_a, w_m, b, g$. Following the recommendations of Radosavovic et al. [38], we constrain our search space by fixing $b$ at 1 and $g$ at 8, reducing

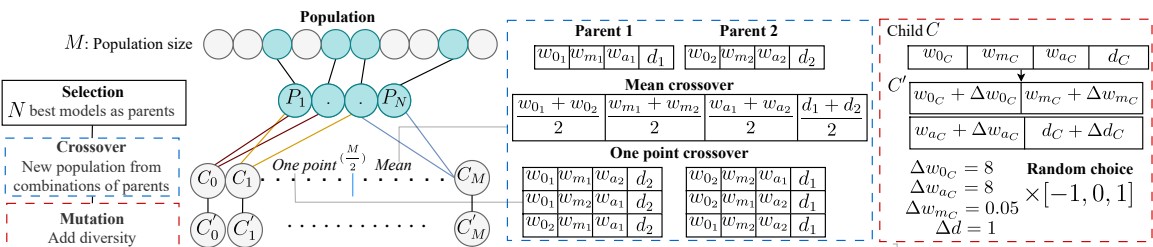

| | | Standard Space | | | Extended Space(+) | | |
|---|---|---|---|---|---|---|---|
| **Parameter** | | **Start** | **End** | **Step** | **Start** | **End** | **Step** |
| $w_0$ | Initial Width | 16 | 120 | 8 | 64 | 240 | 8 |
| $w_a$ | Width Growth Rate | 16 | 64 | 8 | 32 | 128 | 8 |
| $w_m$ | Width Mult. Factor | 2.05 | 2.9 | 0.05 | 2.05 | 2.9 | 0.05 |
| $d$ | Depth | 8 | 22 | 1 | 12 | 25 | 1 |

RegNet quantized linear parameterization

$$u(j) = j \cdot w_a + w_0, \quad j = 0, 1, \dots, d-1$$

$$k(j) = \frac{\log(u(j)/w_0)}{\log(w_m)} \quad w(j) = w_0 \cdot w_m^{k(j)}$$

Quantized per-block widths:

$$w_q(j) = \mathrm{round}\left(\frac{w(j)}{8}\right) \cdot 8$$

Figure 3: **RegNet search space** - Our two proposed search spaces, which are subsets of the RegNet design space, along with the equations that define this design space [38]. We use only the four main parameters that define each RegNet ($d, w_0, w_a, w_m$) to reduce the size of space.

Table 1: Strategies used to enhance the performance of RegNet models, based on the improvements proposed by [39].

| Training methods | Regularization methods | | Architecture improvements |
|---|---|---|---|
| ✓Cosine LR decay
✓Increase training epochs | ✓SWA of weights
✓Label Smoothing
✓Stochastic Depth | ✓RandAugment
✓Dropout on FC layer
✓Decrease weight decay | ✓Squeeze and Excitation
✓ResNet-D |

free parameters from six to four: $d, w_0, w_a, w_m$. By applying the constraints shown in Figure 3, we define two search spaces: a standard version focused on small models with 18,564 possible architectures and an extended version (+) that supports larger models, totaling 58,344 architectures. To boost model performance, we adopted some architectural, regularization, and training strategies originally proposed for ResNets [39] (see Table 1).

## 3.1 Evolutionary Operators

Evolutionary operators guide exploration and exploitation in ENAS, ensuring the discovery of high-performing architectures. Their selection depends on the encoding of architectures and the desired search dynamics. We propose two crossover methods for our RegNet search space, along with mutation strategies to maintain diversity. The breeding pipeline is illustrated in Fig. 4 and will be used with the same parameters for the search step in our experiments.

Figure 4: **Breeding process in EGE-NAS** -. From a population of 20 models, the top 4 by accuracy are selected. One-point crossover generates half of the new population, while mean crossover creates the other half. A similarity score prevents redundancy by applying a mutation if a new individual is too similar to a previously trained one.

## 4 Data augmentation selection based on Zero-cost proxies

To enhance data diversity and thus improve generalization, augmentations can be applied to the dataset. In our case, following the nature of the NAS Unseen Data Challenge [12], we do not know in advance which modality or what information a given dataset contains. However, not all

transformations are effective for every dataset. For example, color and intensity transformations are irrelevant for the LaMelo [14] and Gutenberg [17] datasets, which consist of strings encoded into images without meaningful color information.

We propose a novel method based on zero-cost proxies, as illustrated in Fig. 5a. We trained a RegNetY_400MF model for 50 epochs with 22 candidate augmentations and without augmentation to establish a ground-truth ranking. We then evaluated different zero-cost proxies (one minibatch) with other augmentation methods to assess their alignment with the ground truth across datasets (Fig. 5c). Our results show that the normalized sum of Fisher and Jacob_cov (fisher_jacob) as a ranking metric effectively helps filter out poor augmentations. To ensure model independence, we averaged rankings from 20 random models sampled from the search space in our EG-ENAS. As shown in Figure 5b, our method avoids bad augmentations more effectively than other candidate approaches, completing the selection process in less than 8 minutes per dataset.

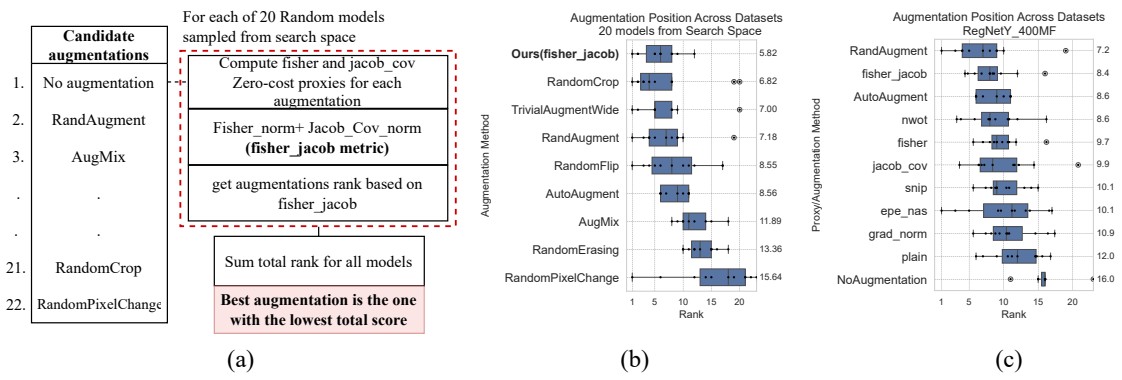

(a)                                    (b)                                    (c)

Figure 5: **Zero-cost proxy-based augmentation selection** – (a) Proposed method: We evaluate 22 augmentations based on Fisher and Jacob Covariance zero-cost proxies. (b) Augmentation rankings across 11 datasets (see Fig. 2), comparing our approach with other augmentation methods. A rank of 1 indicates the best augmentation among the 22 candidates. (c) Augmentation rankings based on zero-cost proxies from a single RegNetY_400MF. Rankings in (b) and (c) are averaged over ten seeds.

## 5 Population initialization

An appropriately selected initial population accelerates convergence of the search process by starting with higher-performing architectures, which is especially beneficial for large datasets or search spaces. However, excessive constraints can reduce diversity, and the optimal population diversity varies by dataset. Common methods used for selecting the initial population include Random Initialization, Search Space Reduction [40], Diversity-based Initialization [41] and Performance-guided Initialization [42].

Taking advantage of the simple encoding of our RegNetY models ($W_0, W_m, W_a, D$), dataset specific metadata (*num_classes, num_channels*) and model metrics (*#params, num_stages*), we propose training a surrogate model that ranks architectures by predicted test accuracy. This regressor does not replace NAS search but filters out weak candidates during the first generation population initialization. The process consists of sampling 100 models from the search space and selecting top 20 based on surrogate model predictions. We selected RandomForest and SGD regressors as surrogate models. To generate training data, we first trained 240 models for each validation dataset for 50 epochs using the strategies listed in Table 1. Instead of predicting absolute test accuracy, we employed pairwise comparisons, resulting in the generation of 149,498 data points.

To validate our regressors, we employ cross-validation using the validation datasets. We rank the models based on the regressors' predictions and then compare the Spearman's rank correlation of our regressors with the correlations derived from rankings obtained after training the models for 5 and 20 epochs, as well as those obtained using the Jacov_cov zero-cost proxy. As shown in Figure 6, both regressors performed similarly and outperformed the ranking correlations based on early training, while making predictions in seconds, with the exception of the Chester dataset.

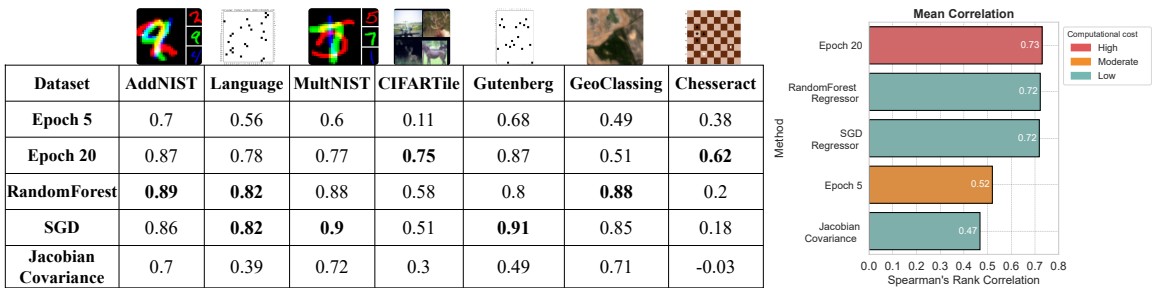

| Dataset | AddNIST | Language | MultNIST | CIFARTile | Gutenberg | GeoClassing | Chesseract |
|---|---|---|---|---|---|---|---|
| Epoch 5 | 0.7 | 0.56 | 0.6 | 0.11 | 0.68 | 0.49 | 0.38 |
| Epoch 20 | 0.87 | 0.78 | 0.77 | **0.75** | 0.87 | 0.51 | **0.62** |
| RandomForest | **0.89** | **0.82** | 0.88 | 0.58 | 0.8 | **0.88** | 0.2 |
| SGD | 0.86 | **0.82** | **0.9** | 0.51 | **0.91** | 0.85 | 0.18 |
| Jacobian Covariance | 0.7 | 0.39 | 0.72 | 0.3 | 0.49 | 0.71 | -0.03 |

Figure 6: **Performance of the surrogate model for population initialization** – Spearman's rank correlation of estimated rankings across 240 models per dataset with different fitness evaluation methods. Our trained RandomForest and SGD regressors achieve correlation scores comparable to training the population for 20 epochs, with significantly lower computational cost.

## 6  Fitness evaluation

One of the main bottlenecks in ENAS is the fitness evaluation step, which involves training and evaluating models to identify the top candidates for breeding the next generation. Our surrogate models help filter out poor-performing models in the first generation, but are not reliable for directly identifying the best ones. As a result, training and evaluation are still necessary, but we aim to minimize the time spent on population training. Common acceleration methods in NAS include Supernet Training [25], Low Fidelity Estimation [27], Partial Training, Meta Learning [43], and Transfer Learning [44]. For our EG-ENAS we employ three key strategies: **Partial Training**, as even training for only 10 epochs provides a strong ranking correlation; **Transfer Learning** and **Weight Inheritance**, reusing pretrained model weights. Weight transferability depends on specific conditions for each part of the RegNet architecture. For the **Stem**, it is applicable only between models with the same input dimensions. For the **Body**, transfer is possible between network stages if both stages have identical widths.

This means that for all NAS generations except the first, we can inherit the Stem weights from the best model found so far during evolutionary search. For the body, which holds most of the network's weights, we created a stage weights pretrained pool using the 1,528 models trained for 50 epochs for our ranking regressor. Based on the quantized linear functions, we developed an algorithm to select the best pretrained model from the pool that maximizes block weight transfer. Figure 7 compares the accuracy of 120 models trained from scratch versus those using our weight transfer and inheritance approach.

## 7  Experiments and results

### 7.1  Experimental setup

We evaluated our EG-ENAS on a NVIDIA A100 GPU using the 11 image classification datasets listed in Figure 2. For comparison with other NAS methods or architectures, we used test accuracy as the primary metric. To aggregate results across multiple datasets, we followed the NAS Unseen

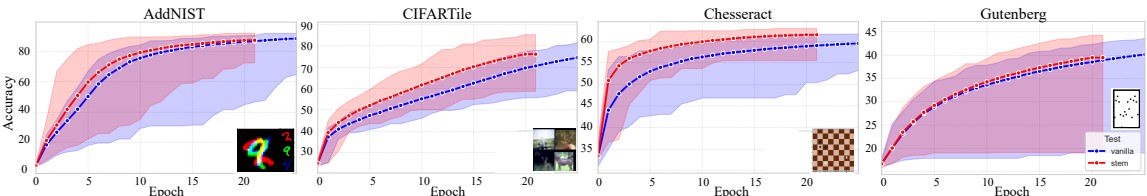

Figure 7: **Validation Accuracy with Weight Transfer and Inheritance** - Validation accuracy of 120 models without stage weight transfer (blue) and with stage weight transfer from a pretrained pool plus stem inheritance from the parent of the last generation (red), evaluated across four datasets. Dotted lines represent the mean population accuracy.

Data Challenge metric [12], which uses a relative score (adj_score) based on benchmark accuracy:

$$\text{scaling factor} = \frac{10}{100 - \text{benchmark}}, \quad \text{adj\_score} = \frac{\text{raw\_score} - \text{benchmark}}{\text{scaling factor}}$$

## 7.2 Ablation studies

To assess the impact of each component of our EG-ENAS on final accuracy and total relative score, we defined **seven training modes**, including two baseline configurations (T1 and T2), covering population initialization, evolutionary computation, and weight sharing/inheritance. Our low-cost mode (T0) uses a surrogate model as a search step to select the best model from 100 random models in the search space. Additionally, we tested five augmentation strategies, naming tests as **Mode + AugmentationStrategy**, with a "+" indicating the use of the extended search space described in section 4. Each test was run with three different seeds. These modes are listed in Table 2. Modes T1 to T7 were tested with 3 generations, a population size of 20 individuals, and training parameters defined in Table 5.

Table 2: **EG-ENAS modes** - Modes used for ablation studies, combining different components of our pipeline (left). Available augmentation selection methods (right).

| Component / Mode | Ranking regressor based initialization | Evolutionary search | Weights transfer and inheritance | Training epochs (search) |
|---|---|---|---|---|
| **T0:** RFR based selection | ✓ | | | 0 |
| **T1:** Random Search | | | | 5 |
| **T2:** Baseline EvoNAS | | ✓ | | 5 |
| **T3:** (RFR initialization) | ✓ | ✓ | | 5 |
| **T4:** (Weights transfer) | | ✓ | ✓ | 5 |
| **T6:** (Full EG-ENAS) | ✓ | ✓ | ✓ | 5 |
| **T7:** (Full EG-ENAS) | ✓ | ✓ | ✓ | 10 |

| Augmentation Selection methods |
|---|
| **AA:** AutoAugment transform |
| **P:** Zero-cost proxies based selection |
| **B:** Basic transform (RandomErasing + RandomCrop + RandomHFlip) |
| **R_10:** Based on ResNet18 model trained for 10 |
| **R_20:** Based on ResNet18 model trained for 20 |

RFR: Random Forest Ranking regressor

## 7.3 Results

The aggregated adj_score and total time for each study, along with error bars from the test seeds, are shown in Figure 8 for both validation and test datasets. We identified two main clusters: the first includes low-cost modes (T0) that took on average less than one hour per dataset. The second cluster includes modes T1-T6, which involve training populations and use the same number of generations, population sizes, and training parameters. On average, they require between 3 and 4 hours per dataset, except for T7, which takes 5 hours. Significant improvements were observed when using the expanded search space (+) in both clusters, particularly for the test datasets. On average, this required just one additional hour per dataset compared to using the standard search space.

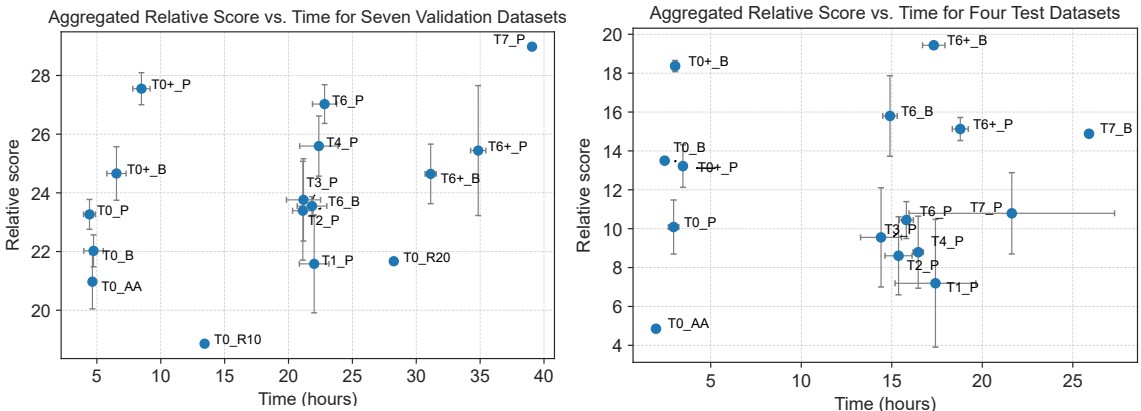

Figure 8: **Relative Score vs. Total Time** - Adjusted total relative score versus total time for each study on validation datasets (left) and test datasets (right). Detailed results for individual datasets can be found in Appendix Figures 11 and 12. Search space size and augmentation selection method were the components that had the greatest impact on the total score for both the Low-Cost mode (T0) and the population training modes (T1–T7).

### 7.3.1 Augmentation selection.
Besides the size of the search space, the augmentation selection method was the component that caused the greatest score differences between the tests. For validation datasets, our zero-cost proxy method (P) outperformed the other augmentation strategies and proved stable across different seeds. For test datasets, the Basic transform (B) was the most effective, with the proxy method in second place. No single transform works best for all datasets. Our augmentation selection method (P) effectively avoids poor augmentations, demonstrating the potential of zero-cost proxies as a novel approach for this task. But, it still fails to consistently identify the best transform that would return the highest scores, highlighting the need for further research on reliable augmentation selection for unseen datasets.

### 7.3.2 Ablation studies.
Our low-cost mode (T0) achieved strong scores comparable to the best competition scores in the NAS Unseen Challenge 2023. This mode requires on average 1 hour per dataset and achieved state-of-the-art results for the Sadie dataset, as shown in Table 3. This highlights the power and potential that surrogate models and zero-cost proxies have in NAS to reduce computation time while maintaining strong performance.

Among the T1-T6 studies, Random Search showed the worst performance(T1), followed by basic ENAS(T2). Using the Regressor for population initialization (T3) and weights transfer(T4) improved the total score compared to basic ENAS(T2), but not as significantly as expected. The best results were achieved by combining both strategies (T6 and T7) across both dataset groups. T7 is the best option for cases where time is not a constraint. The performance of the evolutionary search could potentially be improved by using other parameters like larger population sizes or alternative crossover strategies, which could be explored in future research.

### 7.3.3 Comparison with other methods.
We compare the relative aggregated score and validation dataset test accuracies with scores from several CNN networks, the best score from the 2023 competition (*Best Competition*) and other scores of NAS methods provided by [12]. See Table 3. We also include the mean total time per dataset in seconds. However, since no time data was provided for the CNN or NAS solutions, a direct comparison of efficiency is not possible. To facilitate visualization of the ranking by dataset, Appendix Table 7 shows all methods by dataset, sorted from best to worst. Our method ranks first in the Sadie dataset, achieving a state-of-the-art score. Compared to the best competition scores that require similar computation time, we outperformed them in 4 out of 7 datasets. The results For the test datasets are presented in Table 4. In this case, the scores for

the CNN and NAS models on the Volga [21] and Sokoto [20] datasets have not yet been released. As a result, we are unable to directly compare the performance of our solution. Our work aims, therefore, to serve as a benchmark that encourages researchers to evaluate and compare their approaches on new datasets, aligning with one of the central goals of this paper.

Table 3: **Test Accuracy and Relative Scores for Validation Datasets** - The first section summarizes eight of our studies (mean of three seeds), the second lists various CNN models, and the third compares NAS methods with the top scores from the NAS Unseen Data Challenge 2023 [12]. Note that computation times for these scores are unavailable, limiting efficiency comparisons.

| Method | Mean total(↓) time (seconds) | LaMelo | Gutenberg | Adaline | Chester | Sadie | Mateo | Caitie | Relative Score(↑) |
|---|---|---|---|---|---|---|---|---|---|
| T0_P | 2273 | 87.08 | 47.87 | 95.53 | 61.70 | 96.46 | 94.28 | 72.92 | 25.76 |
| T0+_P | 4364 | 88.2 | 46.07 | 96.73 | 60.41 | **97.09** | 95.71 | 79.05 | 30.14 |
| T1_P | 11319 | 86.97 | 44.52 | 95.89 | 59.35 | 96.44 | 93.96 | 71.38 | 24.28 |
| T2_P | 10862 | 87.09 | 45.2 | 96.21 | 61.28 | 95.69 | 94.39 | 73.92 | 25.82 |
| T6_P | 11736 | 86.55 | 47.67 | 96.53 | 61.04 | 95.78 | 95.82 | 80.83 | 29.05 |
| T6+_P | 17922 | 88.06 | 47.02 | 96.37 | 59.11 | 96.93 | 95.12 | 80.18 | 29.04 |
| ResNext | - | 93.97 | 40.3 | 91.42 | 55.15 | 89.9 | 90.57 | 46.23 | 11.11 |
| ResNet18 | - | **97.0** | 49.98 | 92.08 | 57.83 | 80.33 | 91.55 | 45.56 | 12.16 |
| DenseNet | - | 84.57 | 43.28 | 93.52 | 59.6 | 94.21 | 92.81 | 51.28 | 13.98 |
| MNASNet | - | 84.63 | 38.0 | 90.51 | 56.26 | 86.0 | 87.7 | 48.49 | -0.92 |
| VGG16 | - | 84.54 | 44.0 | 92.06 | 55.69 | 93.67 | 90.43 | 24.43 | 3.77 |
| Best Competition | - | 89.71 | **50.85** | 95.06 | 62.98 | 96.08 | 95.45 | 73.08 | 29.02 |
| Bonsai-Net | - | 87.65 | 48.57 | **97.91** | 60.76 | 95.66 | 97.17 | 91.47 | 34.66 |
| Random Bonsai | - | 76.83 | 29.0 | 34.17 | **68.83** | 63.56 | 39.76 | 24.76 | -37.80 |
| PC-DARTS | - | 90.12 | 49.12 | 96.6 | 57.20 | 94.61 | 96.68 | **92.28** | 33.37 |
| DrNAS | - | 88.55 | 46.62 | 97.06 | 58.24 | 96.03 | **98.1** | 81.08 | 32.75 |
| Random DARTS | - | 90.12 | 47.72 | 97.07 | 59.16 | 95.54 | 96.55 | 90.74 | 34.10 |

Table 4: **Test Accuracy and Relative Scores for Test Datasets** - The first section presents six of our studies (mean of three seeds), the second lists CNN models, and the third includes NAS methods tested on the NASBENCH-201 benchmarks [45]. For NAS methods, the provided time reflects the search time for the best network, while our methods' time includes both search and training time, averaged across the datasets.

| Method | Mean total time (seconds)(↓) | Sokoto | ImageNet16-120 | CIFAR10 | Volga | Relative Score(↑) |
|---|---|---|---|---|---|---|
| T0+_B | 2729 | 89.98 | 45.07 | **94.16** | 83.85 | 19.53 |
| T0+_P | 3118 | 62.57 | 38.93 | 93.70 | 83.90 | 14.40 |
| T1_P | 15671 | 55.58 | 36.77 | 91.91 | 83.57 | 11.10 |
| T2_P | 13836 | 54.78 | 37.27 | 91.66 | 83.78 | 10.86 |
| T6+_B | 15586 | **92.12** | **45.53** | 94.12 | **84.12** | 19.94 |
| T6+_P | 16905 | 75.13 | 44.07 | 93.41 | 83.89 | 16.52 |
| ResNet-18 | - | 28.94 | 46.38 | 93.02 | 71.35 | 0 |
| DenseNet | - | - | - | **95.04** | - | - |
| VGG16 | - | - | - | 92.64 | - | - |
| PC-Darts | - | - | 41.31 | 93.41 | - | - |
| DrNAS | - | - | **46.34** | 94.36 | - | - |
| DiNAS | - | - | 45.41 | **94.37** | - | - |
| NASWOT (N=1000) | 306 | - | 44.44 | 92.96 | - | - |
| ENAS | 13315 | - | 16.32 | 54.3 | - | - |
| REA | 12000 | - | 45.54 | 93.92 | - | - |

## 8 Conclusion and Future Work

The findings of our research offer significant insights for the fields of NAS and AutoML. This study represents the first application of the RegNet search space in NAS, demonstrating its advantages through its simple encoding and high density of effective models. The evaluation of these 9 novel datasets with various augmentations, alongside our zero-cost proxy-based selection method,

highlights the importance of dataset-specific transform selection and demonstrates how zero-cost proxies can quickly indicate augmentation effectiveness. The correlation scores achieved with our ranking regressor, which we employed for population initialization in our ENAS search and for our low-cost method (T0), highlight the advantages of surrogate models as efficient strategies in NAS. By using weight transfer through a pretrained stages pool and parent stem inheritance for RegNets, we achieved a reduction in population training time by 5 to 10 epochs. After evaluating our EG-ENAS with a new and diverse group of datasets, our work advances the state-of-the-art in NAS, while underscoring the need for future research that prioritizes both generalization and efficiency over reliance on common homogeneous image datasets.

We identified the following limitations: We compared final test accuracy scores across datasets with NAS methods and CNN models' scores from [12], but the lack of runtime data made efficiency comparisons difficult. Limited resources in terms of compute also restricted our evaluation of the extended space (+), proxy-based augmentation selection across all test modes (T1-T4), and hyperparameter tuning. Future work could explore other search spaces, such as MobileNetV3, for better comparison and new proxies tailored to specific architectures or search spaces. Further improvements include exploring more robust surrogate models and alternative genetic operators for RegNet models. Lastly, improving weight transfer methods between models, independent of their architectures, could improve NAS efficiency and applicability across diverse datasets and tasks.

## 9 Broader Impact Statement

Our research aims to improve the efficiency and generalizability of evolutionary Neural Architecture Search (NAS) by using surrogate models, zero-cost proxies, and transfer learning. While our method represents an advancement and reduces unnecessary calculations, it still needs a lot of computing power because of its iterative search approach and training. To tackle this, we plan to develop or combine metrics and surrogate models that can adapt to different architectures and datasets. However, we believe that with the work presented here, we are already making a contribution to finding optimized deep learning solutions with less computational effort.

**Acknowledgements**. We receive funding from the Bavarian State Ministry of Science and the Arts (StMWK) and Fonds de recherche du Québec (FRQ) under the Collaborative Bilateral Research Program Bavaria – Québec managed by WKS at Bavarian Research Alliance (BayFOR) and Fonds de recherche du Québec – Santé (FRQS). The presented content is solely the responsibility of the authors and does not necessarily represent the official views of the above funding agencies.

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

# A  Training parameters

The final step of our EG-ENAS involves fully training the selected best model. The training parameters used for this step are shown in Table 5.

```
BATCH_SIZE: 128
EPOCHS: 100
LR: 0.001
MIN_LR: 0.0000
LR_SCHEDULER: "cosine_annealing"
SCHEDULER_EPOCHS: 100
WARMUP: True
LABEL_SMOOTHING: 0.1
WEIGHT_DECAY: 0.01
MOMENTUM: 0.9
TYPE: "Adam"
SWA_START: 90
TOPK: 2
```

Table 5: Training configuration parameters

# B  Candidate augmentations

We trained the 22 candidate augmentations on each validation dataset using a RegNet_400MF model, as explained in Section 4. A value of 0 indicates no augmentation. The full list of augmentation tests is shown in Table 6, and the ranking of each augmentation tested on the seven validation datasets is presented in Figure 9

| ID | Transformations |
|----|-----------------|
| 0  | [] |
| 1  | RandAugment(magnitude=9) if C in [1,3] else RandAugmentMultiChannel() |
| 2  | RandAugment(magnitude=5) |
| 3  | RandAugment(magnitude=1) |
| 4  | TrivialAugmentWide(num_magnitude_bins=31) |
| 5  | TrivialAugmentWide(num_magnitude_bins=15) |
| 6  | AugMix(severity=3) |
| 7  | AugMix(severity=1) |
| 8  | RandomHorizontalFlip(), RandomVerticalFlip() |
| 9  | RandomErasing(p=0.2, scale=(0.05, 0.2), ratio=(0.3, 3.3)), RandomHorizontalFlip(), RandomVerticalFlip() |
| 10 | RandomErasing(p=0.2, scale=(0.05, 0.2), ratio=(0.3, 3.3)) |
| 11 | RandomErasing(p=0.2, scale=(0.02, 0.2), ratio=(0.3, 3.3)), RandomCrop((H, W), padding=(PH, PW)) |
| 12 | RandomCrop((H, W), padding=(PH, PW)) |
| 13 | RandomCrop((H, W), padding=(PH, PW)), RandomHorizontalFlip(), RandomVerticalFlip() |
| 14 | RandomErasing(p=0.2, scale=(0.02, 0.2), ratio=(0.3, 3.3)), RandomCrop((H, W), padding=(PH, PW)), RandomHorizontalFlip() |
| 15 | RandomPixelChange(0.01), ToTensor() |
| 16 | RandomPixelChange(0.025), ToTensor() |
| 17 | RandomPixelChange(0.05), ToTensor() |
| 18 | RandomPixelChange(0.01), ToTensor(), RandomHorizontalFlip(), RandomVerticalFlip() |
| 19 | RandomPixelChange(0.01), ToTensor(), RandomErasing(p=0.2, scale=(0.05, 0.2), ratio=(0.3, 3.3)) |
| 20 | RandomPixelChange(0.01), ToTensor(), RandomCrop((H, W), padding=(PH, PW)) |
| 21 | RandomPixelChange(0.01), ToTensor(), RandomHorizontalFlip(), RandomVerticalFlip(), RandomErasing(p=0.2, scale=(0.05, 0.2), ratio=(0.3, 3.3)) |
| 22 | AutoAugment() |

Table 6: Augmentations tested in section 4

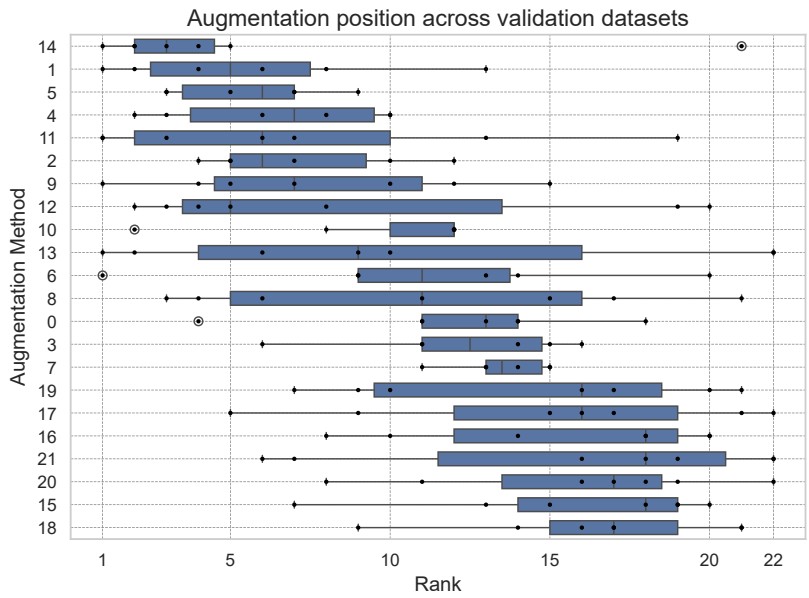

Figure 9: Ranking positions of each augmentation type across validation datasets

## C Ranking Correlation of Zero-Cost Proxies

As an alternative to our Random Forest-based population initialization for EG-NAS, we tested how well different zero-cost proxies ranked a population of models from the RegNet search space. The results are shown in Figure 10.

## D Test accuracies for Individual Datasets

The scores per dataset for our EG-ENAS models are shown in Figures 11 and 12. The position of our EG-ENAS compared to other NAS methods across the seven validation datasets is shown in Table 7.

|     | LaMelo | Gutenberg | Adaline | Chester | Sadie | Mateo | Caitie |
|-----|--------|-----------|---------|---------|-------|-------|--------|
| 1 | ResNet-18 | Best Competition | Bonsai-Net | Random Bonsai | **Ours** | DrNAS | PC-DARTS |
| 2 | ResNext | ResNet-18 | **Ours** | Best Competition | Best Competition | Bonsai-Net | Bonsai-Net |
| 3 | PC-DARTS | PC-DARTS | Random DARTS | **Ours** | DrNAS | PC-DARTS | Random DARTS |
| 4 | Random DARTS | Bonsai-Net | DrNAS | Bonsai-Net | Bonsai-Net | Random DARTS | **Ours** |
| 5 | Best Competition | **Ours** | PC-DARTS | DenseNet | Random DARTS | **Ours** | DrNAS |
| 6 | DrNAS | Random DARTS | Best Competition | Random DARTS | PC-DARTS | Best Competition | Best Competition |
| 7 | **Ours** | DrNAS | DenseNet | DrNAS | DenseNet | DenseNet | DenseNet |
| 8 | Bonsai-Net | VGG16 | ResNet-18 | ResNet-18 | VGG16 | ResNet-18 | MNASNet |
| 9 | MNASNet | DenseNet | VGG16 | PC-DARTS | ResNext | ResNext | ResNext |
| 10 | DenseNet | ResNext | ResNext | MNASNet | MNASNet | VGG16 | ResNet-18 |
| 11 | VGG16 | MNASNet | MNASNet | VGG16 | ResNet-18 | MNASNet | Random Bonsai |
| 12 | Random Bonsai | Random Bonsai | Random Bonsai | ResNext | Random Bonsai | Random Bonsai | VGG16 |

Table 7: Ranking of our best score among NAS methods and CNN models on the validation datasets.

## E Weights transfer in RegNets

The main structure of RegNet and its weight transfer conditions are shown in Figures 13 and 14.

Figure 10: Spearman's rank correlation of the ranking position estimated with 7 different zero cost proxies on 240 models for each dataset.

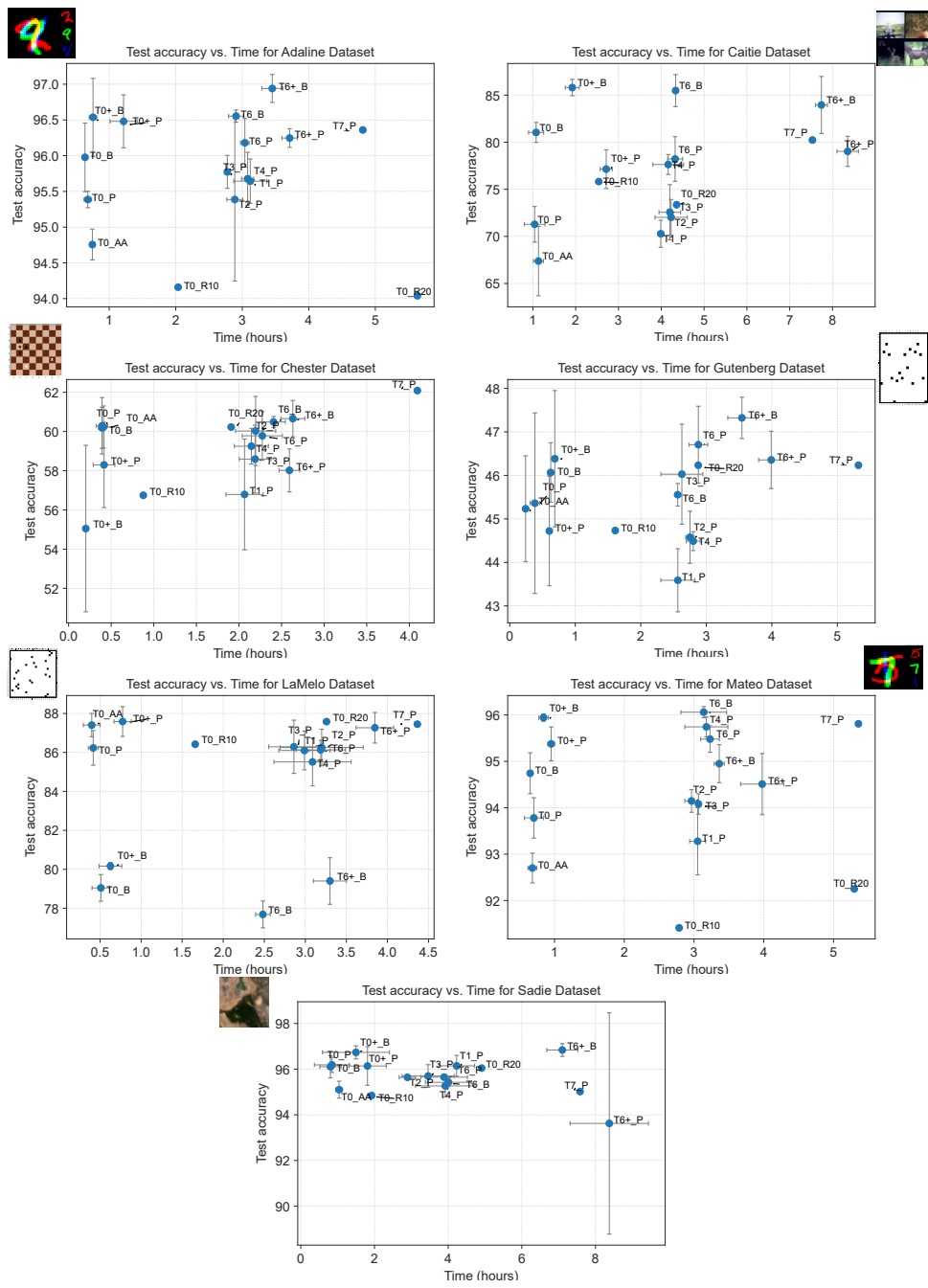

Figure 11: Test accuracy versus time in hours for each study on seven validation datasets

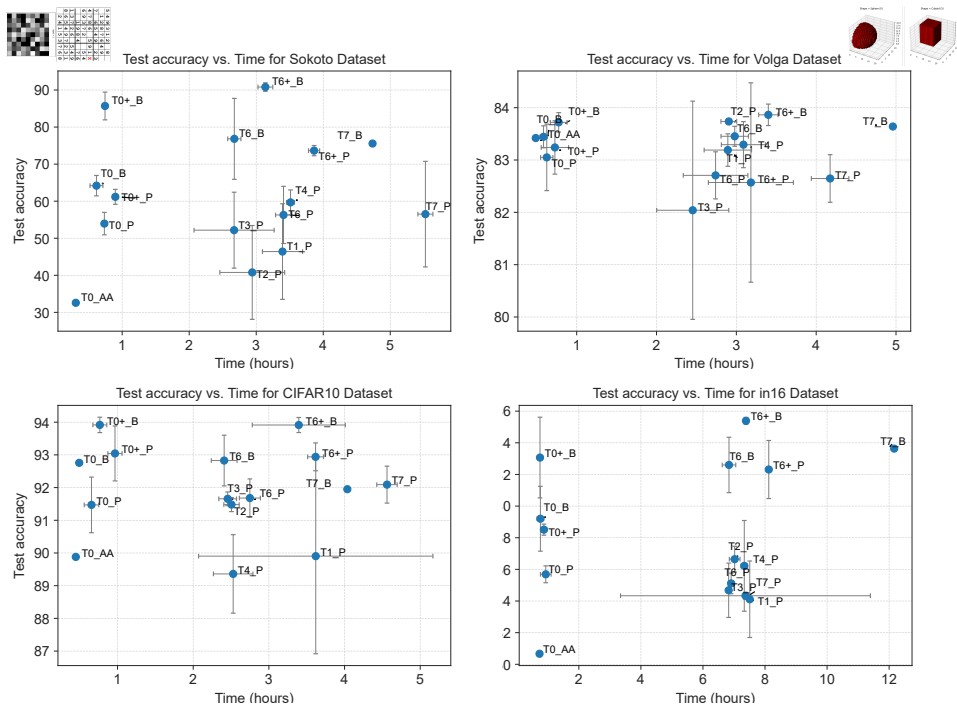

Figure 12: Test accuracy versus time in hours for each study on four test datasets

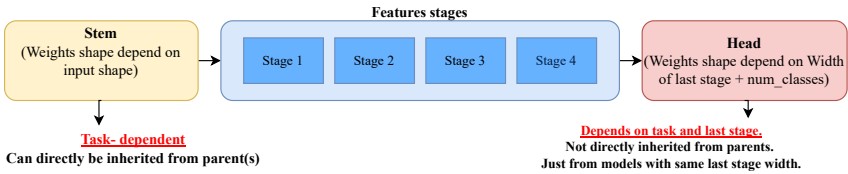

Figure 13: Macro structure of RegNet networks and cases where weights can be transferred to the Stem or Head layers

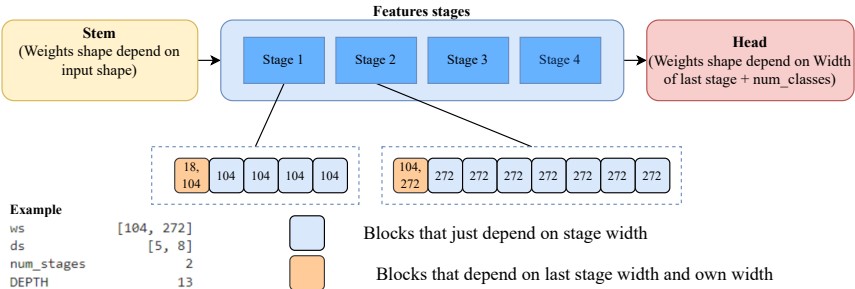

Figure 14: Structure of stages in body of the RegNet networks and cases where blocks can be transferred to the Stages layers

