# OpenReview forum: "EG-ENAS: Efficient and Generalizable Evolutionary Neural Architecture Search for Image Classification"
_automl.cc/AutoML/2025/Methods_Track — AutoML 2025 Methods Track_

### Official Review · Reviewer_E3uS · 2025-04-30

**Comments To Authors:**

Factual Assessment:
- State/summarize the main contributions of the paper in a few sentences.
This paper introduces a novel NAS paradigm based in the RegNetY search space. Within the framework, it utilizes an efficiently pre-trained proxy based on ZCPs, and model rank regressors. This paradigm is so far specific to the domain of image classification, but shows promise over cross-dataset generalization. Finally, they also included a paradigm-specific weight transfer routine that improved model validation accuracy.

- Compare the paper with previous work. In particular, is there highly relevant previously published work that the authors do not seem to be aware of?
The paper references critical previous work discussing the baselines for ENAS, ZCPs, Efficient NAS, and specific works to weight sharing and other used prior research.
- Express your level of confidence in the correctness of the results, and point out any major errors, if any are found.
I am 95% confident in the correctness of the results shown in the paper.

Final assessment:
- What are the strengths of the paper? (results? new research direction? application? etc.)
The strengths of this paper showcase the possibility of combining a plethora of modern NAS and general AutoML techniques to a single paradigm to the benefit of the system as a whole. The paper also performed a variety of useful ablation studies to compare the usefulness of each piece of the proposed paradigm. This alone can be leveraged in future research in order to speed up NAS algorithm training times. The paper also clearly identifies useful prior literature for each technique used within the final paradigm, and leaves necessary training parameters and more in depth results within the appendix.

- What are the weaknesses of the paper?
Weaknesses in the paper include short explanations for a variety of the experimental design for the paradigm. One example is with the design for the model crossover and mutation. While it may be a simple mechanism through which they designed it, there is no explanation within the appendix (section A) as to the function of the system, and no explanation as to the reasoning behind the design.
This also continues withing the discussion of the ablation studies, as each training mode is briefly glossed over, and the reader is left making an educated guess as to the respective training pipeline for each mode. While it would probably be reproducible, it is difficult to guarantee with the information present.

- Express and explain your opinion regarding whether the contributions of the paper (assuming they are correct and original) are interesting/useful/relevant.
This paper identifies an interesting area of further research in complex training paradigms for NAS for image classification.While the scope is rather limited within this paper, it has the capability to be applied to a variety of tasks.

- Final Recommendation: Give a final recommendation for acceptance/rejection (or a more refined distinction, such as borderline).
Approve, with modifications.

I feel this paper could use better explanations for their experimental design, either in the appendix or otherwise, before being approved.

- Additional feedback: Comment on the quality, clarity, and readability of the writing. Provide comments that may help the authors in producing a revised version of the paper.
The paper was clear, and it was straightforward to follow along with the background, results, and specific design of the paradigm.

**Review Confidence:**

4

**Review Rating:**

6

---

### Official Review · Reviewer_mriD · 2025-04-30

**Comments To Authors:**

**Summary:**
This paper aims to improve the efficiency and generalisability of current NAS methods. To achieve this goal, several improvements are proposed to the ENAS algorithm, including reducing the search space size, a dataset-aware augmentation selection method, a ranking regressor to filter low-potential models, and a weight-sharing strategy.

**Strengths:**
1. The ranking regressor model performs very well, achieving correlation scores comparable to training for 20 epochs, with lower computational cost. This result encourages further exploration of ranking-based surrogate models.
2. The augmentation selection method shows notable improvement compared to the baseline, which is particularly helpful when the modality and information contained in the dataset are unknown.

**Weakness:**
1. No time requirements for the competing NAS methods are provided, so there isn't enough evidence to claim that the proposed NAS method is more efficient.
2. The proposed method underperforms compared methods on average across diverse datasets, which does not support the proposed method being more generalisable.
3. Method T1 – T5 only train for 5 epoch during search, and according to Figure 5, this leads to significantly lower correlation compared to ranking regressor, which likely explain the suboptimal performance, replace training during search with ranking surrogate completely could bring potential improvement to T3 - T5 while keeping the efficiency of T0.
4. The search space used in this paper is a significantly reduced version of RegNetY search space, containing only 18 564 (58 344 for extended version) possible architectures, which raises questions about the applicability of the proposed methods to larger search spaces. Is the search space large and expressive enough to be applicable to many types of problems?

**Review Confidence:**

3

**Review Rating:**

5

---

### Official Review · Reviewer_kHQm · 2025-05-01

**Comments To Authors:**

**Summary**:
This paper proposes an evolutionary search approach to enhance generalizability across different datasets. The approach is divided into several steps. The initial step introduces zero-cost proxies to select the most effective data augmentation techniques for each dataset. The second step improves the selection of the initial population of the evolutionary algorithm (EA) by using surrogate models to identify the top 20 networks from a random candidate pool. The last step enhances the search process by leveraging weight-sharing for similar operations, which reduces the number of training epochs required for evaluating the fitness of the networks. The proposed improved EA is then evaluated on the unseen NAS datasets.

**Other works**:
- Current ZCP NAS research uses proxies as input to a performance prediction method for speed-ups in NAS and improved correlation [1,2,3]. This part is missing in this paper.

[1]Kadlecová et al., Surprisingly Strong Performance Prediction with Neural Graph Features, ICML 2024

[2] Akhauri and Abdelfattah, Encodings for Prediction-based Neural Architecture Search, ICML 2024

[3] Lukasik et al., An Evaluation of Zero-Cost Proxies -- from Neural Architecture Performance to Model Robustness, IJCV 2024


**Strengths**:
- This paper provides a strong motivation for the need and importance of improved search approaches for more  the generalizability of NAS
- The proposed approach demonstrates significant improvements over baseline search methods
- Additionally, the study includes comprehensive experiments on diverse datasets, providing a thorough comparison with various methods.


**Weaknesses**:
- Adding more steps to a search seems to improve the EA baseline, but it doesn’t seem to improve against hand-designed networks.
- The search space is only CNN-based. It would be interesting to see if this approach could improve over transformer-based networks.
- The data augmentation steps feel over-engineered, and simple augmentations that could be used for all image tasks could be more effective. Also this selection needs to be done for each dataset a more versatile approach would be better.  In addition, training a RegNetY_400MF on different (in total 22) augmentations in combination with ZCP feels like a heavy additional training step for each dataset.
- The initial population is adapted using surrogates, which first need to be trained. Why not use the ZCPs from the augmentation part instead of training architectures to generate a training dataset for the surrogates?
- Before the search actually starts, several hundred architectures need to be trained, which feels counterintuitive to the goal of reducing training costs in NAS and the paper’s idea. Could these steps not be combined to reduce the pre-training steps, since they need to be done for each dataset?
- A plot of the actual search process would be helpful to understand how well it’s working and which factors are driving the improvements against the baseline EA.


**Additional Comments**:
I feel like this paper aimed to improve the search to make it more generalizable, which eventually led to some over-engineered steps that required additional training before the search could actually begin. A better augmentation could eventually also help existing hand-designed networks, such as ResNet18. However, the biggest issue is that the search didn’t improve the ranking of hand-designed networks, but instead introduced more training and compute time.

**Review Confidence:**

5

**Review Rating:**

4

---

### Meta-Review · Area_Chair_Am87 · 2025-05-07

**Recommendation:** Accept
**Confidence:** 4

**Metareview:**

The paper presents an approach to improve the efficiency and generalizability of evolutionary NAS methods.

Strengths of the paper include reproducible code (for the most part), comprehensive evaluation, and the benefit of the proposed rank regressor, extensive evaluation.

While reviewers kHQm and mriD identified some unaddressed limitations (over-engineered solution,  smaller search space, etc.), in my opinion, the performance of the ranking regressor is appreciable and promising, and encourages further research in this direction. The method also improves performance over many baselines. I therefore recommend acceptance.

I ignored Reviewer E3uS's comments since they were quite generic and not very relevant.